# Hospital Staffing during the COVID-19 Pandemic in Sweden

**DOI:** 10.3390/healthcare10102116

**Published:** 2022-10-21

**Authors:** Ritva Rosenbäck, Björn Lantz, Peter Rosén

**Affiliations:** 1Department of Engineering Science, University West, 461 86 Trollhättan, Sweden; 2Technology Management and Economics, Chalmers University of Technology, 412 96 Gothenburg, Sweden; 3Department of Business Administration, University of Gothenburg, 405 30 Gothenburg, Sweden

**Keywords:** hospital staffing, surge capacity, COVID-19 pandemic, hospital managers, healthcare staffing, mixed method

## Abstract

Staff management challenges in the healthcare system are inherently different during pandemic conditions than under normal circumstances. Surge capacity must be rapidly increased, particularly in the intensive care units (ICU), to handle the increased pressure, without depleting the rest of the system. In addition, sickness or fatigue among the staff can become a critical issue. This study explores the lessons learned by first- and second-line managers in Sweden with regard to staff management during the COVID-19 pandemic. A mixed-methods approach was used, with preliminary qualitative interview (*n* = 38) and principal quantitative questionnaire (*n* = 272) studies, based on principal component and multiple regression analyses. The results revealed that the pandemic created four types of challenges relating to staff management: staff movement within hospitals; addition of external staff; addition of hours for existing staff through overtime and new shift schedules; and avoidance of lost hours due to sickness or fatigue. Furthermore, the effects of these managerial challenges were different in the first wave than in later waves, and they significantly differed between the ICU and other units. Therefore, a greater proactive focus on staff management would be beneficial in future pandemic situations.

## 1. Introduction

The Coronavirus disease 2019 (COVID-19) pandemic has caused a huge global human crisis, with 588 million confirmed cases and >6.4 million deaths, as of 17 August 2022, according to the World Health Organization (WHO)’s recent global report [1]. Sweden had a cumulative number of nearly 2.55 million confirmed cases and 19,500 deaths, according to the local authorities [2]. A total of 9247 new patients were admitted to intensive care units (ICU), with over 500 simultaneous admissions in ICUs in April 2020, during the first wave. The maximum occupation of inpatient beds was approximately 3000 nearly a year later, in January 2021 [2].

The WHO has developed a checklist describing the protocol to be followed during pandemics to ensure citizens’ health and safety, and to concurrently minimize the social and economic disruptions. The WHO report divides a pandemic into different phases for appropriate planning measures. Phase 1–4 represents the initial stages of low infection, which is yet to become a pandemic. Phase 5 describes the spread of the first wave in a few countries, which subsequently increases and spreads worldwide in Phase 6. After Phase 6, a post-peak period occurs with a decrease in the infection rates; this possibly peaks to a second wave, or even more, before the infection acquires the characteristics of a normal seasonal flu in the post-pandemic period [3]. The total duration of a global pandemic is 2–3 years [4]. In Sweden, the first wave started in February 2020, after winter vacation travels to Italy, where the European COVID-19 pandemic had started. The pandemic was in the post-peak period in Sweden during May 2022 and at the end of the fourth infection wave [2].

Currently, there are numerous ongoing studies on COVID-19 worldwide, and noteworthy articles are published almost every week on the approaches adopted by different healthcare settings to manage the sudden influx of patients with COVID-19. Several studies have reported the substantial and rapidly growing burden of the COVID-19 pandemic on hospital resources [5,6,7], including the higher demand for personal protection equipment (PPE), ventilators, oxygen, consumables, inpatient beds, and ICUs managed by knowledgeable staff such as intensivists, anesthetists, and nurses with specialized knowledge. However, adopting new procedures and hygiene protocols for the new patients and preventing the spread of infection has been challenging for the staff. [5,7].

The ability to quickly respond to a sudden increase in demand for healthcare is known as “surge capacity” [8,9]; it is divided into four components or “four S’s”: “staff” (adequate skilled staff), “stuff” (beds and specific equipment needed for patients with COVID-19 such as intubation equipment, mechanical ventilators, and PPE), “structure” (hospitals and facilities for emergency and intensive care), and “system” (management systems that ensure ongoing and proactive planning) [7,10,11]. The “four S’s” have been applied during the COVID-19 pandemic as WHO’s guidance to create surge capacity for acute and intensive care [12]. This study has been limited to exploring the methodology by which Swedish hospitals managed their staffing to swiftly create surge capacity during the COVID-19 pandemic. However, these measures have earlier been reported in the international literature; therefore, they are not considered specific to the Swedish context.

A system’s ability to adapt to a sudden increase in demand during a crisis and subsequently, to quickly recover from it is elucidated by the concept of resilience. However, resilience includes a system’s ability to continuously learn from past experiences of crisis management and improve the ability to manage future crises [13,14]. According to Kruk et al. [15] and Cannedy et al. [16], a resilient healthcare system will safeguard human life and produce good health outcomes during a crisis, and it will revert to its routine of delivering everyday healthcare post-crisis. Sinha and Ola [14] reported the differences between reactive and proactive responses in an organization during crises. Reactive response is the ability to absorb, survive, and reorganize following acute instability, while proactive response is the ability to anticipate and be prepared for possible crises [14,17]. Resilience could be used as the optimal surge capacity response to the wavering needs during a pandemic. O’Connor et al. [18] describe the need for flexible capacity models; for example, an emergency department (ED) may need to have extra staff when the burden is high; however, the excess capacity could be used for non-urgent care after the burden decreases. The COVID-19 pandemic is a unique situation, and O’Connor et al. [18] describe the valued and close inter-disciplinary collaboration when the hospitals were under pressure.

The burden on healthcare organizations has been uneven, for the ICUs in particular, because of the high admission rate of patients and long length of stay of up to several weeks for critically ill patients [5,19]. As the burden increases, certain reduction options may be used, which are partially dependent on the resilience of the network capacity, such as transferring patients, especially ICU patients, to other regions or countries [20], and reduction of elective care, which has been used in many regions and countries with severe infection outbreaks [20,21]. Globally, the burden on healthcare organizations was also reduced, as emergency services were less frequently used during the pandemic [18].

A proactive component of surge capacity is to keep the occupancy rate low (i.e., excess capacity) under normal conditions and utilize this unused capacity during a sudden increase in demand [18]. Sweden, on the contrary, has lowered the number of ICU beds and has the lowest number of hospital beds per 100,000 citizens in Europe [22]. Furthermore, Winkelmann et al. [7] reported the differences between European countries and Sweden, which has a low number of ICU beds (two beds per 100,000 citizens) compared to, for example, Germany, which has eight ICU beds per 100,000 citizens. The highest increase in the number of ICU beds was achieved in Ireland, the Netherlands, and Sweden, where the national capacity doubled compared to pre-COVID levels [5,23]. In France, the ICU-bed capacity was increased by 95% [20] at the beginning of the pandemic. In Spain, the ICU-bed capacity increased by an average of 160%, with the highest increase (four times) at some healthcare institutions [5] Additionally, regarding surge capacity in EDs, in the Netherlands, the treatment capacity was increased by 70% in all EDs, with a median capacity increase of 49% [18].

Globally, the COVID-19 pandemic has strained healthcare systems, and organizations in different nations and regions have had different approaches to expanding surge capacity. Therefore, it is important to take advantage of the lessons learned by healthcare managers to manage future pandemics. Thus, this study aimed to explore the lessons learned by first- and second-line managers in ICU and other units (ED and inpatient care units) on the approaches Swedish hospitals applied to expand staff surge capacity during the first and later waves of the COVID-19 pandemic.

This study contributes to the literature on surge capacity in healthcare by developing new empirical knowledge through a mixed-methods study, with data from the Swedish healthcare system, on the approaches adopted by Swedish hospitals to expand surge capacity, including the differences in the approaches used between the first wave and later waves of the COVID-19 pandemic and between different types of care units (ICU, ED, and inpatient care units). Furthermore, we examined the satisfaction among first- and second-line managers at ICU and other units with the approach of the hospital in managing staffing during the first and later waves. To the best of our knowledge, no previous studies have systematically identified and analyzed different approaches to enhance surge capacity in healthcare systems using this holistic methodology.

This study is organized as follows: Section 2 provides a brief literature review on the different approaches to dealing with a sudden influx of patients; Section 3 presents the research methodology; Section 4 and Section 5 present the empirical results from the qualitative and quantitative studies, respectively; Section 6 discusses the results; and Section 7 concludes the study with our inferences, managerial implications, and suggestions for future research.

## 2. Literature Review

This section summarizes the results of our literature review on different approaches to expanding surge capacity in three categories: internally and externally added staff, added hours, and lost hours. The following keywords were used alone or in combination to search for relevant literature in Scopus and Google Scholar: healthcare, staff, staffing, surge capacity, and COVID-19.

### 2.1. Internally and Externally Added Staff

The added staff were intended to care for patients, which allowed the specialized staff to treat critically ill patients and train and/or act as consultants to supervise the added staff [5,24,25]. The regular unit staff’s responsibilities could be extended; for example, intensivists in Spanish hospitals could treat 3–4 patients instead of 1–2 in normal situations [5].

During the first wave of the pandemic, the focus was on increasing the healthcare workforce, mainly in the newly established COVID-19 departments in hospitals. This included the recruitment of new staff, such as physicians and nursing students, recent graduates, specialized knowledgeable retirees, inactive healthcare workers, military physicians, staff from private healthcare providers, staff from less affected regions, and foreign healthcare workers. In addition, volunteers were enrolled for simpler tasks, such as arranging food and cleaning PPE for the other staff [7,21,24,26]. According to Winkelmann et al. [7], the most common strategy in European countries was to recruit medical and nursing students to support health care practitioners, in addition to extending the working hours of the existing workforce.

When elective care is reduced, the burden on different units at the hospital will be uneven. This makes it possible to move staff (voluntary or imposed) from other departments of the hospital. Previous studies have reported staff movements within the hospital, such as physicians and nurses from different specialties assisting in the ICU, ED, and inpatient care units [7,21,24,27]. In the Netherlands, 61% of the EDs were expanded with nursing staff and physicians from other departments for COVID-19 emergency care [18]. Melman et al. [6] report that in the UK, operation theater staff could augment the ICU because of their expertise. Additionally, anesthesiologists could knowledgeably contribute to the ICU requirements [20]. However, Jensen et al. [27] have reported that many of the healthcare workers who had moved perceived that their new job did not resemble their usual job at all, or only resembled it to a small degree. Consequently, the staff in the new positions needed rapid training. Brickman et al. [28] have reported the importance of training and its management, when it is required by many, and have examined the training methodology in New York, wherein a 3 h training course allowing nurses to work alongside experienced critical care nurses was implemented. Training was provided to several hundred nurses in a span of 10 days at the beginning of 2020.

Healthcare organizations have, throughout the COVID-19 pandemic, used both voluntarism and imposition when relocating staff. A study conducted in Denmark by Jensen et al. [27] reported that 57% of the study participants re-located from their regular job, and 80% of these were imposed or partially imposed to their new position. Chow et al. [29] and Jensen et al. [27] suggested that imposed staff are less satisfied and have less training experience than others, while voluntarism could lead to a positive experience of professional growth, also in addition to self-growth, societal awareness, and leadership and team management skills [29].

Relocations that occur voluntarily are important for staff well-being, due to concerns about family safety and the availability of PPE [5]. Chow [29] suggests that personal narratives and role modeling by senior staff can be invaluable in recruiting volunteers. Practical factors, such as childcare support when schools are closed, ensuring transportation to reach the workplace when public transport was shut down, and providing additional housing to protect family members encouraged staff to perform their duties admirably [7,27].

### 2.2. Added Hours

A common approach to enhance surge capacity is to increase the working hours of the existing professional healthcare workers by enforcing overtime, modifying work schedules, canceling leaves of absence, transitioning from part-time to full-time work, and increasing the work shifts [5,7,26]. Nunez-Villaveiran et al. [5] reported that some Spanish ICUs combined 8- and 12 h shifts, depending on the number of ICU patients.

Increasing the working hours of existing healthcare workers was a common approach to create surge capacity in healthcare systems during the COVID-19 pandemic [7,26]. However, extra workload under unfavorable circumstances significantly increases stress among healthcare professionals, which can lead to sick leave and lost working hours [7]. Therefore, Dichter et al. [30] have proposed limiting overtime to <50% of normal working hours to minimize the risk of burnout and fatigue.

### 2.3. Lost Hours

The number of staff needed is dependent on the number of absent staff. The staff could be absent due to infection or isolation due to symptoms, awaiting analysis results, infection in family, psychological, or other physical factors [7]. Pendharkar et al. [31] have reported, based on experience, that 25% of absent physicians were included as a buffer in their pandemic plan. During the COVID-19 pandemic, some European countries reported a three-fold higher risk of infection among healthcare professionals than the public due to a lack of PPE [7]. This may be due to differences in the severity of the community transmission and shortage of PPE or the staff’s inadequate understanding regarding the correct use of the PPE [21].

The risk of infection or carrying the infection and death in the family was stressful for the staff, and several studies have reported the mental impact of the risk at work [7,27,32]. Jensen et al. [27] reported that 89%, 36%, and 40% of the professionals were apprehensive of transmitting the infection to their families, colleagues, and patients, respectively. Therefore, it is crucial to protect healthcare workers by ensuring the availability of PPE and training them in its correct usage [31,33].

Several studies have reported increased psychological issues such as stress, anxiety, or frustration during the pandemic, due to not feeling knowledgeable enough about the work, fatigue or burnout caused by heavy workloads, and observing significant numbers of deaths. This has created an awareness among the management of the need to protect the staff’s mental health and well-being, such as the need to be absent, rest, and recover [7,21,27,32,34,35,36]. Hawaei et al. [37] studied various aspects of nurses’ work environment in Canada during the COVID-19 pandemic and their impact on nurses’ post-traumatic stress disorder, anxiety, depression, and emotional exhaustion. The authors reported that workplace safety, access to resources and supplies, organizational support, and workplace relationships were the most protective of nurses’ mental health.

## 3. Materials and Methods

### 3.1. Design

To develop a broad understanding of the staffing situation at Swedish hospitals during the COVID-19 pandemic, a mixed methods approach was employed. The methodology follows one of Morgan’s [38] priority models for mixed methods, wherein a qualitative study delivers preliminary results to develop content for the questionnaire of a quantitative survey. An inductive methodological approach was used to analyze data from a total of 36 semi-structured interviews from 2 cases, based on questions on the methods of finding surge capacity during the different waves of the COVID-19 pandemic [39]; for example, “What elements were the most important for building the right capacity during the COVID-19 pandemic”. The results provided an overview of the situation and its management, which was followed up via a web-based questionnaire for managers in ICU, inpatient care units, and EDs in hospitals all over Sweden to validate the findings of the qualitative study. Both the qualitative and the quantitative study were broader, and only the data about how to build staff surge capacity are used in this study.

### 3.2. Data Collection

Sweden has a total of 7 central region hospitals and approximately 70 other hospitals in 21 regions. Two different cases were chosen to obtain comprehensive information and extend the emergent theory [39]. Case A was a medium-sized hospital in a region with a large population, which had an early and severe outbreak of COVID-19. Case B was a central region hospital in a small region in terms of population, which was less exposed to infection. Both regions had >2 hospitals, which helped each other, and patients were transported between the hospitals during the pandemic.

To ensure adequate sample representation in the qualitative study, all managers who were declared as those managing COVID-19 units were interviewed. Case A included interviews with a total of 27 respondents (note that two of the interviews were with two respondents): members of the hospital management group including the CEO; CMO (chief medical officer); managers of operations, human resources, communication, education, and care; managers of ED, inpatient care units, and ICU; and service managers handling logistics and maintenance. Case B was the central hospital in the region, and some of the departments were organized as regional departments; hence, the chosen management was slightly different. In total, 11 interviews were conducted with the regional director of health and medical care; the manager and coordinator of the staff from the regions’ crisis management center; the manager of the local hospital’s special crisis management center; managers of the ED, the department of infectious diseases, internal medicine, and service; the chief physician; and the chief hygienist physician. The manager of the ICU in Case B was unavailable. The researchers episodically conducted the interviews to ensure that the interviewees spoke more freely [40], and an interview guide was used as memory support to prevent missing any points. The interview guide consisted of 5 groups of questions, and the ones pertaining to this study were questions on building staff capacity, both “who” and “how” to recruit extra staff, and “how to take care of the staff during a pandemic.” All interviews were recorded, transcribed, and subsequently used for data analysis.

The web-based questionnaire, developed from the results of the qualitative study, consisted of background questions, such as “which region of Sweden”, “what type of department”, and “which type of employees they managed.” Additionally, there were 9 groups of independent items, with an overall total of 58 items. The groups of items used in this study comprised items on “where the extra staff came from”, and in case of internal staff, if they moved voluntarily or not (8 items); items on “how increased working hours for the unit staff was managed” (4 items); and items on “how the staff strength was reduced due to illness” (4 items). The questionnaire was completed by 6 items of dependent items, such as if the respondent perceived that the hospital managed the situation in their department admirably and their satisfaction with the management’s handling of the situation. The item “I am satisfied with how the hospital managed the staffing in the first wave” repeated for the later waves, was used for the results in this study. All items, except for the background item, were divided into items A and B, which queried the situation in the first and later waves, respectively. A five-point Likert scale was used to record answers for each item, where a lower value meant a lower level of agreement with the statement. The questionnaire was tested on two experts in healthcare organizations and then on five potential respondents in the target population before distribution, and after minor adjustments, it was sent to the target population.

All 21 Swedish regions were requested to participate in the quantitative study. The e-mail addresses of all first- and second-line managers of ICU, inpatient care, and EDs, caring for patients with COVID-19 at each hospital in the regions and/or hospitals, were sought via their official mailbox for all except one, for whom the e-mail address was unknown. Some regions and hospitals sent plenty of unsorted e-mail addresses; a few others shared the second-line managers’ e-mail addresses for direct contact with the first-line managers; some requested the personnel’s permission and sent the e-mail addresses thereafter, and some others denied provision of access due to time shortages. In total, we received e-mail addresses for 773 first- and second-line managers of ICU, inpatient care units, and EDs.

### 3.3. Non-Response Analysis in the Quantitative Study

We received a response rate of 35.2% (272 of 773), which is considered average in this type of study [41]. High response rates are known to reduce the risks of non-response bias [42]; however, these risks need to be assessed. The 4 most applied techniques used to assess non-response bias in survey-based research are (1) extrapolation based on the assumption that late respondents are similar to non-respondents; (2) comparing respondents to the population on characteristics known a priori; (3) comparing respondents to non-respondents on characteristics known a priori; and (4) sampling non-respondents [43]. The lack of information rendered technique (3) unfeasible; hence, techniques (1), (2), and (4) were employed in this study, considering that non-response bias was unlikely to be an issue. First, the analysis based on extrapolation followed the traditional pattern of comparing the sample units who responded before they received the second reminder with those who did not respond until after, based on the assumption that late respondents are more similar to non-respondents. The 2 groups were compared using Mann–Whitney’s U-test, based on all individual Likert items in the questionnaire. The number of significant tests was consistent with the expected number of type I errors. Second, the sample distribution of the type of staff the respondents managed was mainly consistent with the actual distribution at the population level. Third, non-respondents were sampled and queried about their reason for not responding to the initial questionnaire, even after being repeatedly reminded. The characteristics of the answers received implied that bias was not the motive for non-participation; typical reasons were lack of time, not being a manager of a relevant type, and that the questionnaire was categorized as in the e-mail system’s spam filter. To summarize, we proceeded under the assumption that non-response can be regarded as random in this study.

### 3.4. Data Analysis

The interview data from the qualitative research were inductively coded using the software program NVivo for an in-depth understanding of data [44]. First, the order codes were developed, and after discussions between the researchers, these were grouped into themes such as “how the staffing was achieved” and if the extra staff moved voluntarily or not, to theoretically describe the observed phenomena [40].

The questionnaire data were empirically analyzed due to the inductive approach. Initially, the differences between sub-items were descriptively analyzed. After that, Principal Component Analysis (PCA) with Varimax rotation was used to further explore these data. Bartlett’s test of sphericity and the Kaiser–Meyer–Olkin (KMO) measure of sampling adequacy were used to evaluate the PCAs. This was followed by multiple regression analyses to investigate how the managers’ satisfaction with the hospital’s management of staffing in different situations related to different approaches to expanding surge capacity. Backward elimination was used, with successive removal of independent variables based on the criterion probability of F-to-remove < 0.100, consistent with the otherwise exploratory approach in this study [45].

### 3.5. Ethical Considerations

This study did not include medical research involving human subjects, material, tissues, or data, nor did it process or include sensitive or any other type of personal data, nor does it meet any other similar ethical criteria according to Swedish regulations. Hence, a formal ethical review of the study was not required [46,47].

## 4. Results: Qualitative Study

In the qualitative study, the staff capacity needs during the COVID-19 pandemic in EDs, inpatient units and ICUs were examined [19].

The staff capacity needs in the EDs and inpatient care units were partly covered within the units due to decreased patient flow of other emergency patients and reduction of elective care, and they were partly covered by other units [19]. The ED demand increased due to the need to separate the flows of infected patients and other patients, but decreased due to the shorter care time caused by easier hospitalization. The ICU differs from the other units due to their greater need for surge capacity. In Case A, for example, the ICU raised its capacity during the peak of the infection waves to >300% and needed extra staff, both physicians and nurses, preferably with experience in intensive care.

The codes developed from data of the transcribed interviews on the staff were grouped into the following themes:Staff relocated from other parts of the hospital, imposed or voluntary (internally added staff).External recruitment of staff: healthcare educated or not (externally added staff).Change of hours due to overtime, longer shift, double shift, or fewer hours for other tasks (added hours).Staff reduction due to COVID-19 infection or other illnesses (lost hours).

### 4.1. Internally Added Staff

Staff from other parts of the hospital, especially from units treating elective patients, whose workload were reduced, were moved to the units that needed more capacity; for example, EDs, inpatient care units treating patients with COVID-19, and the unit with the greatest need, which was the ICU. The replacement staff, in all cases, were untrained in the tasks that were required to take care of these patients before the pandemic.

The cases solved the need for surge capacity in different ways. Initially, Case A used imposed replacements to increase the capacity of existing units taking care of the rising volume of patients infected with COVID-19. During the first wave, the staff found the need to move reasonable, though the movements were often stressful; later, the movements became voluntary and therefore less stressful. In Case B, the setting up of a new infection unit staffed with voluntary replacements was attempted; however, due to operational issues, staff were imposed to existing units pinpointed to care for patients with COVID-19.

Both physicians and nurses were needed in the ED, and during the first wave, staff were moved to the department; however, during later waves, the ED in both cases had less help from others, mostly due to their own choice because the new staff members were stressed and worried, and this disturbed their work. It was possible to manage without the extra staff due to the decreased flow of emergency patients, and additional empty beds in the inpatient care units; hence, the average care time at the ED was lower than normal.

The pressure on the ICU during the COVID-19 pandemic was high at the hospitals in both cases. Nurse anesthetists, who are nearly as knowledgeable as the intensive care nurses, were the first to move to intensive care in both cases. Although they lacked special intensive care training, they moved to the ICU without any instructions during the first wave, in Case A. Moreover, surgical nurses assisted in the ICU during the first wave; however, when replacements became voluntary in Case A, many of them chose not to return to the ICU. This unevenly distributed the pressure, because the nurse anesthetists had to manage both the ICU and operation theater for prioritized patients.

Both cases had special crisis agreements with the union, with the possibility of compel staff to move within the hospital, with or without special compensation. At the beginning of the COVID-19 pandemic, these agreements were untested and unknown, leading to intense discussions between the management, staff, and unions. Relocating staff, who needed to change daytime work to shift work, was a particularly unpopular issue in both cases, and it happened often because elective treatment is usually conducted during the daytime and working weeks.

### 4.2. Externally Added Staff

In both cases, there was a need for extra staff from outside the hospital, especially in Case A, which was situated in a region with higher community transmission. Both cases managed to employ a few external healthcare-educated staff during the pandemic.

In Case A, the managers were planning for a scenario in which one ICU nurse was required to take care of three patients with the help of assistant nurses and other non-healthcare-educated assistants. The managers advertised through newspapers and social media and employed a high number of assistants with no healthcare experience; for example, retrenched staff from a nearby industry or workers from a circus with no audience. These external staff were educated to be ICU assistants to help with menial work, allowing the nurses to have more time for the patients. Initially, the ICU assistants in Case A were numerous, because they could have been needed if the situation had been exacerbated.

In Case A’s region, although there are several private healthcare providers, there were no agreements that required them to supply staff to government hospitals during a crisis. However, during the COVID-19 pandemic, private employers permitted their staff to take up employment at government hospitals. A few of these staff moved to the bigger hospital in the region. However, Case A did not have any nurses or physicians from private healthcare providers. In Case B’s region, there were no private healthcare providers.

### 4.3. Added Hours

The knowledgeable staff of ICU, infection, and EDs worked hard in both cases during the COVID-19 pandemic. The management used overtime; longer shifts, up to 12.5 h; and double shifts. Double shifts were often used when the non-staffed shift occurred because the manager was unable to find staff willing to work, and due to short-term absences, such as sick leave. Managers at all levels worked overtime, especially at the beginning of the pandemic.

### 4.4. Lost Hours

During surge capacity building for the care of patients with COVID-19, the decrease in staff was high in both cases due to infection, fatigue, and physical illness; being in quarantine or having childcare issues were other important factors that challenged the management of the units. The quarantine rules were stringently enforced by the authorities and during most of the COVID-19 pandemic, individuals or those with family members showing even minor signs of infection were required to stay home.

Several full-time staff were infected by COVID-19, acquired mostly from colleagues, family, and society, and less from the patients. Between waves 2 and 3, 65% of one unit’s staff at the surgery department in Case A was infected with COVID-19. Case B had a later and smaller first breakout; the manager in charge of infection prevention foresaw the problems of colleagues infecting each other and expeditiously decided to limit the number of staff in the staff rooms.

Work during the pandemic has been extremely stressful. The staff had to work with an unknown infection and care for severely ill and dying patients, and this has affected them. Initially, managers in Case A supported the staff through group or individual reflections. First-line managers attempted to have open reflections after each day of work. The HR manager in Case A regularly participated in the daily meetings to ascertain where help was needed, and directed the chaplains and psychologists to those requiring counseling. In Case B, similar crisis support was established by the HR unit to tend to the staff’s worries; additionally, the hospital church provided support.

At the time of conducting the interviews, the COVID-19 pandemic had lasted for more than a year and several of the interviewed managers spoke about the staffs’ and their own fatigue. During the first wave, more of a “fighting spirit” was present among the staff; however, the long-enduring pandemic that has resulted in plenty of work and overtime for the employees has left them exhausted.

### 4.5. Conclusions from the Qualitative Study and Development of the Questionnaire

The qualitative study explores the complicated work involved in staffing the COVID-19 healthcare centers in the ED, inpatient units, and the ICU, and the results were the starting point for the quantitative study. Staffing was conducted differently from the normal procedure and with or without voluntary work, and the methods could affect the staff’s well-being. The dropout of staff due to sickness, stress, or fatigue is an important factor in surge capacity building. The quantitative study aimed to look further into the spread of the problem in Sweden and to validate the findings in the qualitative study, through a web-based questionnaire.

Table 1 describes the developed items from the qualitative study; these statements were used both for the first and later waves. The items for the first wave are numbered 1–17, and the later waves 18–32. Number 33 (first wave) and 34 (later waves) are statements of dependent character.

The extra staff moved to the needy units were from within the hospital and were both voluntary or imposed, and items 1 and 2 (Table 1) queried if the managers procured internal staff and if their move was voluntary during the first wave. Items 3–8 inquired about other methods of staffing in the COVID-19 patient care units during the first wave. The qualitative study has observed that the staff in the units treating patients with COVID-19 needed to work more due to needing their specialized knowledge and items 9–12 mentioned the different methods to handle this issue. Items 13–15 pertained to the reduction of staff due to infection, psychological, or other physiological illnesses. Item 16 pertains to the reduction of staff due to the authorities’ recommendation; for example, pregnant staff, who should not care for patients with COVID-19. Item 33 is the respondents’ perception of the management’s handling of the situation, and this is considered a dependent item.

The longitudinal change during the pandemic is noteworthy; therefore, all items in Table 1 have been repeated for the later waves. The ICU was different due to the higher demand for surge capacity compared to ED and other units, and it was therefore analyzed separately in the quantitative study. This result from the qualitative study anticipated a difference in the experiences between the ICU and other units. This difference was remarkable enough for further analysis; therefore, questions on the type of unit the respondent worked at were included in the questionnaire’s beginning.

## 5. Results from the Questionnaire

Table 2 presents the descriptive statistics for all 34 questionnaire items. Items 1–32 have been sorted pairwise for easier comparison between the pandemic’s first and later waves. Additionally, Table 2 presents the *p*-values from the t-tests where the difference in mean value between each pair (items 1 and 17, items 2 and 18, etc.) was separately examined for the ICU and other units. Some, though not all such pairs, were characterized by a significant difference. In general, the differences were representations of a more difficult situation during the first wave than during later waves, in the ICU and other units. Furthermore, Table 2 presents the *p*-values from the t-tests where the difference in mean value between the ICU and other units was examined for each item. The overall pattern of the differences is that some aspects regarding the staffing situation were more difficult in the ICU than in other units during the first wave, and the typical persistence of this effect during later waves can be observed.

Four PCAs were conducted to further explore these data; two on items 1–16 (one for ICU and another for the other units) and two on items 17–32 (one for ICU and another for the other units), aiming to identify a lower number of underlying factors. Bartlett’s test of sphericity was significant (*p* < 0.001) in all 4 cases; however, the Kaiser–Meyer–Olkin (KMO) measure of sampling adequacy varied between 0.533 and 0.728, indicating that the data may be unsuitable for PCA [48]. Moreover, the factor loadings were generally low, and the reliability values (Cronbach’s α) for the resulting factors were unsatisfactory, which, in conjunction with the sample sizes that were low for PCA, led to the analyses based on single items.

A multiple linear regression analysis with items 1–16 and item 33 as independent and dependent variables, respectively, was conducted to analyze the managers’ satisfaction with the hospital’s management of staffing in the ICU during the first wave. A final model, with three explanatory variables, was established (item 10 was excluded from this analysis because, as indicated in Table 2, it had no variation). The final model with three explanatory variables is exhibited in Table 3, which reveals that items 2 (“Staff from other units served voluntarily on our unit during the first wave”), 3 (“We recruited properly trained staff during the first wave”), and 15, (“Staff were on sick leave due to physical illness other than COVID-19 during the first wave”) all had a significantly positive association with item 33 (“I am satisfied with the way the hospital managed the staffing during the first wave”).

A similar multiple linear regression analysis with items 1–16 and item 33 as independent and dependent variables, respectively, was conducted to analyze the managers’ satisfaction with the hospital’s management of staffing in other units during the first wave. A final model, with two explanatory variables, as exhibited in Table 3, was established. Table 3 reveals that item 2 (“Staff from other units served voluntarily on our unit during the first wave”) had a significantly positive association with item 33 (“I am satisfied with the way the hospital managed the staffing during the first wave”), and item 9 (“We worked more overtime than in a normal period during the first wave”) had a significantly negative association with item 33.

Subsequently, a multiple linear regression analysis with items 17–32 and item 34 as independent and dependent variables, respectively, was conducted to analyze the managers’ satisfaction with the hospital’s management of staffing in the ICU during later waves. A final model, with three explanatory variables, as exhibited in Table 3, was established. Table 3 reveals that item 25 (“We worked more overtime than in a normal period during later waves”) had a significantly negative association with item 34 (“I am satisfied with the way the hospital managed the staffing during later waves”), while items 18 (“Staff from other units served voluntarily on our unit during later waves”) and 28 (“Longer work shifts were used during later waves”) had a significantly positive association with item 34.

Finally, a multiple linear regression analysis with items 17–32 and item 34 as independent and dependent variables, respectively, was conducted to analyze the managers’ satisfaction with the hospital’s management of staffing in other units during later waves. A final model with four explanatory variables, as exhibited in Table 3, was established. Table 3 reveals that items 23 (“Staff from external private caregivers served on our unit during later waves”), 24 (“Volunteers served on our unit during later waves”), 25 (“We worked more overtime than in a normal period during later waves”), and 31 (“Staff were on sick leave due to physical illness other than COVID-19 during later waves”), all had a significantly negative association with item 34 (“I am satisfied with the way the hospital managed the staffing during later waves”).

In addition, it was noted that the variance inflation factor (VIF) values were low for all regression models, indicating that multicollinearity was not an issue in any of the regression models.

## 6. Discussion

The results reveal that the pandemic generated four types of challenges for staff management. These includes addition of internal staff by moving staff within the hospital (imposed or voluntary); addition of external staff (healthcare educated or not); addition of hours for existing staff (overtime and new shift schedules); and avoidance of lost hours due to sickness or fatigue.

These managerial challenges transformed during the pandemic and were different in the first wave than in the later waves, in line with previous studies [3,4,18], and they differed significantly between ICU, which had higher demands during the pandemic, and other units. Sweden’s low amount of both inpatient and ICU beds made the need for surge capacity immediate in the first wave, caused by a high occupancy rate from the beginning [22,23]. In later waves, surge capacity was still needed, as the conditions were stressful for the staff and increased psychological issues such as fatigue, anxiety, and burnout. Protecting the staff’s mental health and well-being then became a challenge [32,36,37].

### 6.1. Internally Added Staff

The existing staff were inadequate to manage the ICU. Regression analysis revealed that recruitment of properly trained staff during the first wave was positively associated with the degree of satisfaction managers experienced with staffing management in the ICU; however, this was only during the first wave. This is consistent with reports from other researchers [7,21,24] that knowledge is highly valued in healthcare organizations. A physician or nurse who is knowledgeable in intensive care could independently work with severely ill patients and, additionally, supervise unacquainted staff. At the beginning of the pandemic, hospitals and society were scoured for staff with intensive care knowledge; however, an average of <2.5 (Table 1; Items 3 and 19) reveals that recruiting knowledgeable staff was challenging.

The second-best staff were employees with other specialist knowledge compared to the regular staff in the needy units. The analysis revealed that voluntarily serving staff from other units was positively associated with the degree of satisfaction managers experienced during the first wave with staffing management in the ICU and other units, which was more available due to decreased demand at other units delivering elective care; for example, outpatient or elective surgery units. The anesthesia staff in this study were invaluable for the ICU, similar to previous reports [6,20]. Addition of imposed or voluntarily moved staff (3.75 and 3.42, respectively, in the ICU, and 2.95 and 2.58, respectively, in other units) was graded highest in the quantitative research for both the ICU and other units. In the qualitative study it was found that individual training for the new tasks is of importance to feel safe, in line with previous studies [28]

Both the ICU and other units had significantly lower average results concerning imposed and voluntarily moved staff during the later waves. This decrease could be because of the reduction in the sense of urgency due to the continuing waves of the pandemic and due to stress. The moved staff felt affected by their lack of knowledge; this is consistent with Jensen et al.’s report [27]. The stress affected both the moved staff and additionally, the ICU staff, who sometimes felt it better to manage by themselves rather than to take care of the stressed moved staff. An example is the ED manager in Case A, who said that they attempted to cope with the situation using normal staff due to the stress caused by other staff. This was found in Case B as well. Daily management meetings facilitated late decisions on the need for help, and subsequently, remaining understaffed was considered safe.

The cases differed in and shifted between having imposed or voluntary movements of staff. Case A had imposed staff movements initially, and later shifted to voluntarism because of the stress felt by the staff when imposed upon. Conversely, Case B had more voluntary movements of staff initially and discussions on the staffing for a new infection unit, and subsequently, the management determined that the number of staff in every unit not treating patients with COVID-19 should move to the needy units. Consequently, a calmer situation was experienced, including a more even stress distribution. The view of uneven distribution with voluntarism than imposition is not revealed in the literature study where voluntarism is lifted as the best procedure [27,29].

In Sweden most decisions are consensus based; hence, during the later waves, moving the staff was difficult due to the lack of consensus. However, as the qualitative research demonstrated, more voluntarisms resulted in unbalanced pressure on staff; for example, when the surgical nurses stayed at the operation theater with fewer tasks due to the reduced elective care instead of helping at the ICU. This is as stated in a quote from the quantitative study’s free text:

“We have not imposed many to go to the COVID-care, but we have tried to solve it more voluntarily. However, I ask myself if it is always the right way to go? If we had had tougher strategies maybe we would have been able to distribute the pressure in a better way. For some periods of time, many have been underemployed when staff in the medical inpatient care have been struggling.”

### 6.2. Externally Added Staff

Regression analysis results revealed that service by external private caregiving staff was negatively associated with the degree of satisfaction managers experienced with staffing management in other units during later waves. Qualitative analysis of Case A indicated that a few of these staff had gone to the bigger university hospital in the region. Private caregivers are often mainstreamed with knowledge of few types of tasks; thus, when they are recruited to a higher level of hospital, their knowledge may be insufficient. These caregivers were not considerably employed during the later waves or at other units during the first wave.

Service by volunteers was negatively associated with the degree of satisfaction managers experienced with staffing management in other units during later waves. Volunteers have inadequate knowledge about healthcare and could only perform routine tasks, including managing PPE storage and lunch arrangements, among others, similar to those reported in previous studies [7,21,24,26]. They often could not participate in patient care. In Case A, volunteers within the ICU helped with patient care, which was necessary; however, there were negative comments on crowding around the beds. In the other units, which did not need much help during later waves, the volunteers were probably unoccupied.

### 6.3. Added Hours

Surge capacity with knowledgeable staff is gained via a higher number of hours from the units’ own specialized staff. The highest graded score for all groups was to avoid education and conferences (ICU, 5.0 and other units, 4.89), while the second highest was to work overtime (ICU, 4.53 and other units 4.21); both of these are consistent with earlier research [5,7,26]. Furthermore, both were significantly higher in the ICU and not significantly decreased during the second wave or later, which is different from the aforementioned added staff score. More overtime was negatively associated with the degree of satisfaction managers experienced with the staffing management in the ICU during later waves, and in other units during the first and later waves. The use of a large amount of overtime is probably because the management has not resolved the situation.

There were fewer longer work shifts than overtime in the ICU, both in the first and later waves; however, it was positively associated with the degree of satisfaction the managers experienced with staffing management in the ICU during later waves. Longer work shifts often had more free time. Many units, in both cases, preferred longer shifts, which meant sharing one day in two 12.5 h shifts rather than working normal shifts and ad hoc overtime during the pandemic. Longer shifts were significantly less employed at other units, probably because of a lesser need for surge capacity.

### 6.4. Lost Hours

Lost hours due to infection and psychological and other physical illnesses considerably impact surge capacity. All three types of items on illnesses were significantly higher in both the ICU and other units during later waves than in the first wave. This might indicate that the staff became fatigued later in the pandemic, due to overwork and plenty of overtime with big responsibilities. The organizations appeared insufficient in supporting the staff, as also seen in previous research [25,30].

During later waves, the respondents graded an average of >3 that the staff’s COVID-19 infection affected the organization, possibly due to the more infectious variant in the second wave. The effect of the infection on the organization was graded by the ICU as significantly lower than in other units. This could be because the ICU was prioritized and received better PPE from the beginning, due to the aerosol-forming tasks during intensive care; moreover, it could be because the ICU staff worked more and spent less time at home, in society, or in coffee rooms, which were the common places for acquiring infection.

Staff on sick leave due to physical illness other than COVID-19 was positively associated with the degree of satisfaction managers experienced with staffing management in the ICU during the first wave. The staff’s physical sickness (other than COVID-19) had low averages overall; however, the lowest was <2 in the ICU during the first wave. This positive association may be due to the staff’s motivation to work in the ICU during the first wave, even if they had other physical illnesses, and helped reduce the work for others. Conversely, in other units, there were indications, for example from Case A, that people stayed home when there was a risk or push for moving them to the needy units; hence, physical illness had a significantly higher grade in other units than in the ICU.

The pandemic has placed immense pressure on staff, both in their own surge capacity-requiring unit or when moved to a needy unit. Most study participants [27] found that the move to a new unit had few similarities to their normal jobs. This includes hospital location, type of medical profession, and imposition of relocation as the main variables associated with the training and movement to a new unit. Staff and units are dissimilar. Intensive care specialists treat unconscious, severely ill patients who require complete care. Emergency staff prefer to anticipate the unknown and be prepared for the consequences. Inpatient care staff’s focus is to ensure that patients recover sufficiently to be discharged home, while outpatient care staff help patients on a daily basis. The resilience of staff from different units is varied. The more resilient staff cope more effectively with stress, overcome challenges more optimistically, have lower PTSD symptoms, and are less anxious [32,34,35,36,37]. ICU staff experienced extremely stressful situations; hence, after the first wave and during later waves, sick leaves due to exhaustion or mental health issues were significantly higher than in other units.

Respondents from all groups highly graded the item on the decrease in capacity caused by transferring staff from risk groups to non-pandemic-related work. This recommendation from the authorities was followed. Notably, it is apparent that the need for support from colleagues during the pandemic outweighs the need for isolation to reduce the risk of infection. A free text from the questionnaire is quoted below:

“We sat too crowded because we did not have any alternative, to get colleagues to walk far away to take a coffee and to sit by themselves was not possible. The most important support and discharge that everybody got when taking a break and socializing with colleagues.”

In the literature, surge capacity [8,9,10,11,12] and resilience are discussed [13,14,15,18] as possible ways to manage a crisis and especially a pandemic [14,17]. Sinha and Ola [14] and Chamberland-Rowe et al. [17] reported the differences between proactive and reactive management for organizations to be able to be resilient. In this article, we go further into the concepts in the conclusion.

## 7. Conclusions

The COVID-19 pandemic caught both people and healthcare systems unawares, including hospitals in Sweden. The need for surge capacity in healthcare was enormous, especially for knowledgeable ICU staff. The pandemic has long endured, and the most knowledgeable staff have been consistently working hard, sometimes while near exhaustion. Help from other units was inadequate, and those staff were additionally exhausted by the stress of moving. This high risk, both for staff and patients, should aid in gaining knowledge on learning to cope with future pandemics.

The Swedish hospital management became reactive during the COVID-19 pandemic. Our research reveals that proactive management, such as previously scheduled longer shifts, are preferable to reactive management, such as overtime.

Proactive training was preferred rather than reactive training in the first working shift at the new unit, with possibly a large number of patients and few staff.

This proactivity would enhance preparedness for future pandemics; for example, having a low occupation rate, which will cause a higher number of knowledgeable staff to be “diluted”, caused by the need for surge capacity.

Moreover, if staff in the elective units could be proactively educated in the requirements of the ICUs, this will possibly ease the pressure on these units and staff during a pandemic. This type of proactive management of advance pandemic placements or cooperation with internal pools is not found in the literature. Both these actions would reduce the uneven pressure on the hospital staff and stress levels of both parties and increase willingness to move and contribute to the surge capacity.

Education and training for contributing to other units need to be individualized according to the acquired and required knowledge. This special training should include work tasks that are meaningful both for the moved and resident staff. This indicates that the staff in every unit that might need additional capacity during future pandemics should divide their tasks into groups; for example, tasks that could be handled by outpatient care nurses or those that could be handled by non-healthcare-educated staff.

Furthermore, proactiveness could mean having prepared routines for decreasing elective care, the contributions of external healthcare providers, possible ways to change shifts and overtime and their compensation, psychological support for the staff, and aiding staff in family care, social support, and childcare. These routines for a pandemic need to be different from advanced routines, such as during emergencies.

Research in which the first- and second-line management are questioned is essential in order to understand the support required by the units for building surge capacity. In this study, we have investigated staff; however, when preparing for a pandemic, the other 3 “S´s” (“stuff”, “space”, and “system”) are equally crucial. Therefore, future research should investigate these components, which in turn would aid with the preparation of pandemic contingency plans.

This study is characterized by several limitations. One limitation is that a retrospective questionnaire-based study, with anonymous and different response rates in different regions, could have been influenced by factors such as overweighing results from higher-rates regions. In addition, no psychometric analysis of the survey was conducted, due to the exploratory orientation of this study. Furthermore, this study was conducted in Sweden; hence, it might not be generalizable to other countries’ healthcare systems. Therefore, the results should be cautiously viewed.

## Figures and Tables

**Table 1 healthcare-10-02116-t001:** The items developed for the questionnaire.

Item	Statement
1	Staff from other units were imposed to serve on our unit during the first wave
2	Staff from other units served voluntarily on our unit during the first wave
3	We recruited properly trained staff during the first wave
4	We recruited staff without proper training during the first wave
5	We hired staff through external temporary employment agencies during the first wave
6	We hired staff through internal staff pools during the first wave
7	Staff from external private caregivers served on our unit during the first wave
8	Volunteers served on our unit during the first wave
9	We worked more overtime than in a normal period during the first wave
10	We canceled courses and conferences with no relation to COVID-19 during the first wave
11	We utilized extra work teams/on-call lines during the first wave
12	Longer work shifts were used during the first wave
13	The staff’s own illness in COVID-19 affected operations during the first wave
14	Staff were on sick leave due to exhaustion or mental health issues during the first wave
15	Staff were on sick leave due to physical illness other than COVID-19 during the first wave
16	Staff in groups at risk were transferred and did not work with COVID-19 patients during the first wave
33	I am satisfied with the way the hospital managed the staffing during the first wave

**Table 2 healthcare-10-02116-t002:** Descriptive statistics and pairwise comparisons.

													ICU vs. Other Units
Item	Mean	S.D.	Median	IQR	*n*	*p*	Mean	S.D.	Median	IQR	*n*	*p*	*p*
1	3.75	1.50	4	3	84	0.001	2.95	1.68	3	4	166	0.001	<0.001
17	2.95	1.51	3	3	84		2.37	1.58	2	3	163		0.006
2	3.42	1.27	4	1.25	84	0.017	2.58	1.44	2	3	166	0.004	<0.001
18	2.93	1.34	3	2	84		2.13	1.35	2	2	159		<0.001
3	2.49	1.43	2	3	82	0.865	2.44	1.38	2	2	172	0.749	0.782
19	2.45	1.24	2	2	84		2.39	1.32	2	2	172		0.715
4	1.62	1.11	1	1	81	0.271	1.75	1.23	1	1	167	0.169	0.418
20	1.45	0.77	1	1	82		1.57	1.13	1	0	165		0.394
5	2.43	1.52	2	3	84	0.608	2.40	1.50	2	3	172	0.154	0.869
21	2.55	1.48	2	3	84		2.63	1.54	3	3	168		0.682
6	2.36	1.51	2	3	84	0.876	2.45	1.58	2	3	163	0.862	0.665
22	2.32	1.45	2	3	84		2.48	1.57	2	3	161		0.448
7	2.14	1.40	1	2	81	0.003	1.44	1.02	1	0	166	0.003	<0.001
23	1.56	1.01	1	1	81		1.16	0.62	1	0	164		<0.001
8	1.30	0.88	1	0	82	0.315	1.22	0.68	1	0	165	0.179	0.431
24	1.18	0.68	1	0	83		1.13	0.50	1	0	163		0.552
9	4.53	1.04	5	0	85	0.389	4.21	1.19	5	1	177	0.912	0.035
25	4.65	0.70	5	0	85		4.22	1.15	5	1	175		0.002
10	5.00	0.00	5	0	85	0.001	4.89	0.48	5	0	181	0.002	0.036
26	4.82	0.47	5	0	85		4.70	0.66	5	0	179		0.118
11	4.67	0.94	5	0	85	0.386	3.88	1.57	5	2	166	0.098	<0.001
27	4.55	0.90	5	1	84		3.59	1.63	4	3	165		<0.001
12	2.89	1.89	2	4	84	0.901	1.99	1.55	1	2	174	0.972	<0.001
28	2.86	1.82	3	4	84		2.00	1.50	1	2	173		<0.001
13	2.49	1.29	2	2	83	0.006	3.45	1.37	3	3	181	0.026	<0.001
29	3.06	1.34	3	2	81		3.76	1.23	4	2	179		<0.001
14	2.18	1.07	2	2	82	<0.001	2.10	1.11	2	2	179	<0.001	0.574
30	2.88	1.16	3	2	82		2.54	1.21	2	1.75	178		0.034
15	1.84	0.81	2	1	79	0.002	2.14	1.04	2	2	169	0.126	0.024
31	2.29	1.01	2	1	80		2.32	1.11	2	2	168		0.848
16	3.90	1.36	5	2	82	0.721	3.88	1.38	5	2	169	0.868	0.885
32	3.83	1.26	4	2	82		3.85	1.33	4	2	168		0.901
33	2.65	1.32	2	2	85	0.420	2.77	1.35	3	3	171	0.557	0.484
34	2.81	1.34	3	2	85		2.86	1.35	3	2	169		0.796

**Table 3 healthcare-10-02116-t003:** The final regression analysis results of the managers’ satisfaction with the hospital’s management of staffing.

	During the First Wave						During Later Waves					
	in the ICU			in Other Units			in the ICU			in Other Units		
Item	B	S.E.	*p*	B	S.E.	*p*	B	S.E.	*p*	B	S.E.	*p*
(Constant)	0.883	0.554	0.115	3.443	0.481	<0.001	3.399	1.069	0.002	5.799	0.570	<0.001
2	0.198	0.116	0.094	0.173	0.082	0.027						
3	0.216	0.104	0.042									
9				−0.268	0.101	0.010						
15	0.322	0.187	0.089									
18							0.312	0.118	0.011			
23										−0.278	0.163	0.092
24										−0.422	0.197	0.035
25							−0.410	0.225	0.073	−0.325	0.115	0.006
28							0.148	0.083	0.080			
31										−0.292	0.107	0.007
F	4.120			5.270			4.482			7.508		
*p*	0.010			0.007			0.006			<0.001		
R^2^	0.162			0.087			0.174			0.236		

## Data Availability

Data presented in this study are available on request from the corresponding author. The data are not publicly available due to further ongoing analysis and publication.

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
