# Peer review of "Hospital Staffing during the COVID-19 Pandemic in Sweden"

_healthcare, 2022, doi:10.3390/healthcare10102116_

Round 1
Reviewer 1 Report
The manuscript address an important subject of staffing during Covi19 crisis. But unfortunately the manuscript is not appropriate for publishing because of the following problems:
1- The objective of the study is not clearly defined and a level of ambiguity is seen in the manuscript that is because the objectives of the study were not clear.
2- The manuscript consists of three parts, first a literature review, second a qualitative study including designing a questionnaire, and finally a quantitative study. Unfortunately the methodology of each part has not been described adequately. For example the search strategy of the first part is not clear. The key questions and rigor of qualitative part is not described, and the psychometric analysis of the designed questionnaire has not been provided, and data gathering of quantitative part has not been presented adequately.
3- The developed questionnaire doesn't seem to cover all the aspects of the subject.
4- Data analysis and result section doesn't provide useful information about the subject.
Reviewer 2 Report
Thank you for the opportunity to review your manuscript. The paper presents an interesting topic. I have no comments on the quality of the review. Please supplement the methodology with a flow chart that clearly and legibly presents the investigation carried out related to the bibliographic search.
Regards!
Reviewer 3 Report
This article focuses on the staff management at hospital during the Covid-19 pandemic in Sweden.
For this purpose, a cross-sectional study with a mixed-methods approach (a qualitative and a quantitative study) was performed to study the factors that enhance surge capacity in healthcare system.
The introduction and the literature review clearly state the problem and the materials and methods section is clear about the design. Nevertheless, several elements need to be clarified:
There are 34 interviews reported, but there are 27 in Case A and 11 in Case B.
There are 8, 6 and 4 items mentioned in the questionnaire, respectively, and only 16 in Table 1.
There is no indication of the scale used to answer the questions (Likert scale...?) nor in which direction the values go (does the best satisfaction correspond to the highest or the lowest value?).
The paragraph on the analysis of non-responses, on the other hand, is very well detailed.
The results section of the qualitative study is clear and well written.
In the results section of the quantitative section, there are several areas of improvement:
Some paragraphs concern more the method part than the results part (use of Principal Component Analysis, Bartlett’s test, backward elimination, which covariates are independent or dependent).
Some information is not necessary (number of iterations and t) and others not explicit (Std. Beta).
The results would be more readable if all four models (Tables 3, 4, 5 and 6) were combined into one table.
The verification of the absence of multicollinearity is on the other hand a positive point.
The discussion summarizes the results well.
Round 2
Reviewer 1 Report
This manuscript address an important issue but unfortunately the authors have tried to include an extensive and wide range of methods in this manuscript that had made it not scientific reliable without details of every part. The qualitative part is not sound and the experiences can not be seen on it. The development of the questionnaire and its psychometric properties are not clear. The main findings that have some helpful information can be presented in a short communication.
Author Response
Thank you for your comment. In line with the decision of the Guest Editor, we have not made any further changes to our manuscript as a result of this comment. Instead, we have responded to the Guest Editor’s comments separately.
Reviewer 3 Report
All my comments have been taken into account and and this article seems to me quite correct to be published as is.
Author Response
Thank you very much for your nice comment.